# Perinatal outcomes of symptomatic chikungunya, dengue and Zika infection during pregnancy in Brazil: a registry-based cohort study

Thiago Cerqueira-Silva [1,2] ✉, Laura C. Rodrigues[1], Neil Pearce [1], Maria Gloria Teixeira[3], Maria da Conceição Nascimento Costa[3], Luciana Cardim[3], Viviane S. Boaventura [2,4], Deborah A. Lawlor [5], Mauricio L. Barreto [3] & Enny S. Paixao [1,3]

The previous literature shows mixed conclusions regarding the risk of adverse perinatal outcomes in pregnant women with symptomatic chikungunya, dengue, and Zika. We investigated this topic using a linked population-based Brazilian cohort from 2015 to 2020. The study included 6,993,395 live births. Among these, 6066 ( < 0.1%) mothers were notified with chikungunya during pregnancy, 19,022 (0.3%) with dengue, and 8396 (0.1%) with Zika. Symptomatic maternal chikungunya was associated with an increased risk of preterm birth (Hazard ratio: 1.10, 95%CI 1.01-1.22), low Apgar score 5' (1.44, 1.14-1.82), and neonatal death (1.50, 1.15-1.96). Symptomatic maternal dengue was associated with preterm birth (1.07, 1.02-1.12), low birth weight (1.10, 1.04-1.15), congenital anomalies (1.19, 1.03-1.37), and low Apgar score 5' (1.26, 1.09-1.45). Symptomatic maternal Zika was associated with all adverse birth outcomes, particularly congenital anomalies, which were over twice the risk (2.36; 1.91-2.67) compared to the unexposed group. This study provides evidence of the adverse consequences of arbovirus infections during pregnancy, including critical time windows by trimester. Our findings emphasise the importance of implementing effective measures to prevent chikungunya, dengue, and Zika infections during pregnancy and the associated adverse birth and neonatal outcomes, which may have long-term health consequences for mothers and their children.

The global burden of arthropod-borne viral (arbovirus) diseases has substantially increased in recent years. Approximately 50% of the global population resides in regions susceptible to transmitting chikungunya, dengue, or Zika viruses[1]. In 2024, there were more than 7.6 million cases of dengue[1], 620,000 cases of chikungunya[2], and nearly 24,000 cases of Zika[3] worldwide.

It has been proposed that the adverse foetal outcomes of arbovirus infections during pregnancy are primarily mediated by mechanisms

[1]Faculty of Epidemiology and Population Health, London School of Hygiene & Tropical Medicine, London, UK. [2]Laboratório de Medicina e Saúde Pública de Precisão—Fundação Oswaldo Cruz, Salvador, Brazil. [3]Centro de Integração de Dados e Conhecimentos para Saúde (Cidacs), Fundação Oswaldo Cruz, Salvador, Brazil. [4]Faculdade de Medicina da Bahia, Universidade Federal da Bahia, Salvador, Brazil. [5]MRC Integrative Epidemiology Unit, University of Bristol, Bristol, UK. ✉e-mail: thiago.silva@lshtm.ac.uk

involving maternal immune activation and placental damage[4–6]. The secretion and circulation of immune-related factors, such as cytokines, may stimulate uterine contractions, resulting in preterm delivery[5]. Chikungunya, Dengue, and Zika viruses can infect the placenta, causing inflammation and endothelial dysfunction that disrupt normal placental function and compromise foetal development[5,7].

All three infections can be vertically transmitted (2–48%)[6,8,9], and this is particularly common during the third trimester for Chikungunya and Dengue viruses. Vertical transmission of Chikungunya and Dengue viruses can present with a disease that resembles neonatal sepsis, leading to severe complications, such as neurodevelopmental delay[10,11]. Vertical transmission of Zika mainly affects the foetus during the first trimester through direct impact on neurogenesis[6,12].

Few epidemiological studies have investigated the clinical consequences of arbovirus infection during pregnancy on maternal, birth and neonatal outcomes, and the existing literature has yielded mixed results[4,13–16]. While the consequences of prenatal exposure to the Zika virus on offspring's neurological development are well documented[12,16,17], other pregnancy-related outcomes have received less attention. The existing literature is primarily based on small cohorts (fewer than 100 pregnancies exposed to Zika), with some studies indicating an increased risk of small for gestational age, although this is not consistently observed[6,16,18,19]. For dengue, there is a more extensive body of evidence; in particular, a meta-analysis that included 36 studies with nearly 40,000 pregnancies exposed to the Dengue virus found an increased risk of neonatal deaths but no

difference in preterm birth and low birth weight, with considerable heterogeneity among the results of each study[13]. The few studies on chikungunya are limited in sample size[8,20–22], with the largest study evaluating 658 pregnancies infected during pregnancy, making it difficult to evaluate outcomes accurately[23].

The studies of these three infections to date have lacked power to explore multiple outcomes and whether the associations differ by the timing of the infection. Also, these studies have methodological flaws, including not classifying and analysing the exposure as time-varying, which can result in immortal time bias, misclassifying unexposed time as exposed, and potentially producing spurious protective associations or biasing the effect estimates downwards[24,25].

This study aims to explore the potential associations of three arboviruses (chikungunya, dengue and Zika) on clinically relevant adverse birth and neonatal outcomes: preterm birth, low birth weight, small for gestational age, large for gestational age, occurrence of any congenital anomaly, low Apgar 5' score and neonatal death, using data from Brazil, where all three viruses are endemic. To our knowledge, this is the largest study to date, and it has better control for critical confounders, appropriately addresses immortal time bias and has investigated the associations of infection by trimester of exposure.

## Results

The study included 6,993,395 live births from 2015 to 2020. (Fig. 1) Among these, 6066 ( < 0.1%) were registered with chikungunya, 19,022 (0.3%) with dengue, and 8,396 (0.1%) with Zika. (Table 1) The maternal

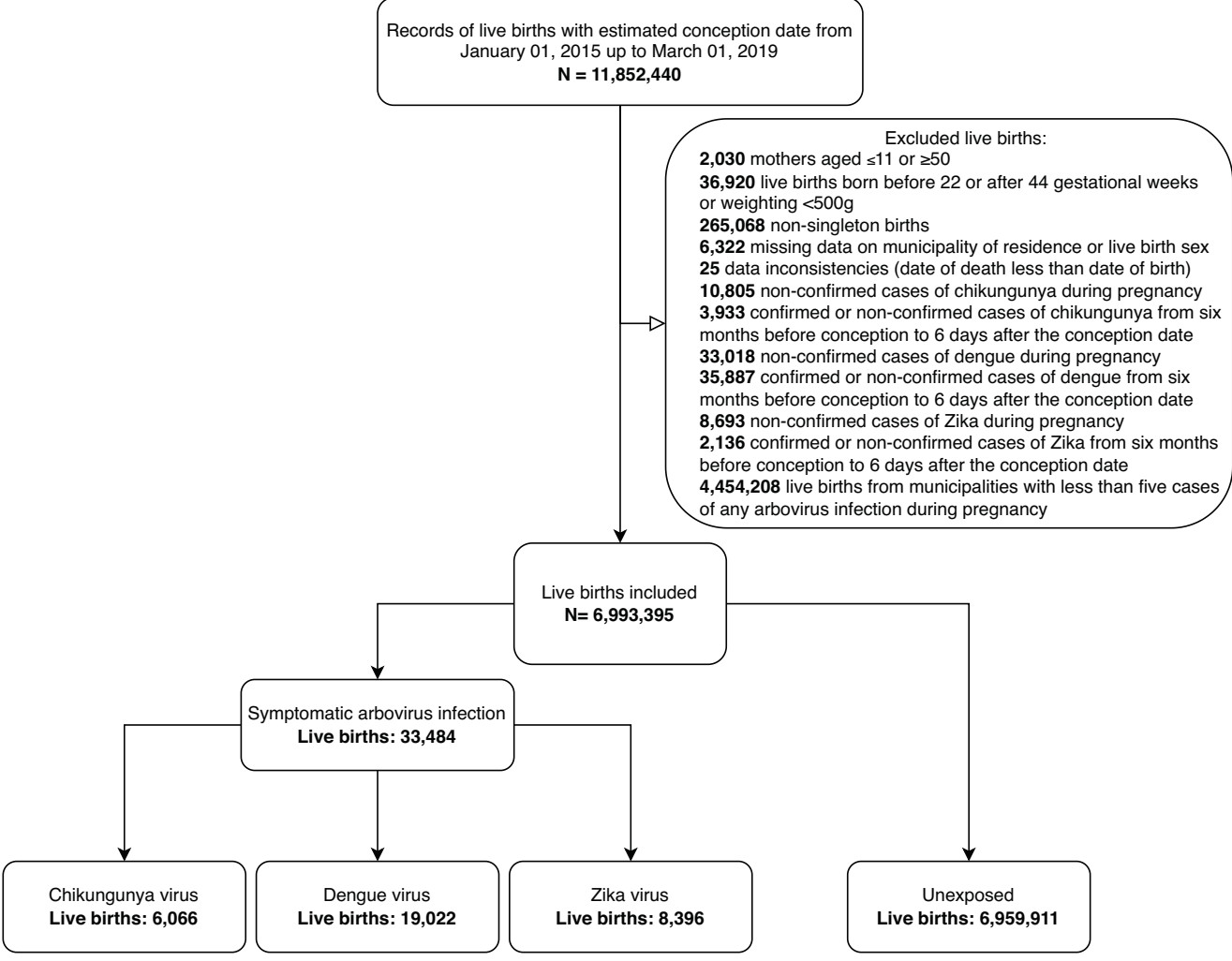

**Fig. 1** | Selection of study participants.

## Table 1 | Baseline characteristics of singleton live births

| Characteristic | Unexposed, N = 6,959,911 | Chikungunya, N = 6066 | Dengue, N = 19,022 | Zika, N = 8396 | Overall, N = 6,993,395 |
|---|---|---|---|---|---|
| **Mother characteristics** | | | | | |
| **Age (years)—mother, median (IQR)** | 27 (22–32) | 26 (21, 31) | 26 (21, 31) | 27 (22, 32) | 27 (22, 32) |
| **Geographic region of residence** | | | | | |
| North | 636,595 (9.1%) | 411 (6.8%) | 680 (3.6%) | 717 (8.5%) | 638,403 (9.1%) |
| Northeast | 1,715,931 (24.7%) | 3,970 (65.4%) | 3705 (19.5%) | 796 (9.5%) | 1,724,402 (24.7%) |
| Southeast | 3,501,294 (50.3%) | 1439 (23.7%) | 10,201 (53.6%) | 5472 (65.2%) | 3,518,406 (50.3%) |
| South | 322,083 (4.6%) | 0 (0.0%) | 575 (3.0%) | 15 (0.2%) | 322,673 (4.6%) |
| Central west | 784,008 (11.3%) | 246 (4.1%) | 3861 (20.3%) | 1396 (16.6%) | 789,511 (11.3%) |
| **Race/ethnicity** | | | | | |
| White | 2,249,165 (32.3%) | 815 (13.4%) | 5871 (30.9%) | 2623 (31.2%) | 2,258,474 (32.3%) |
| Black | 449,834 (6.5%) | 230 (3.8%) | 1023 (5.4%) | 384 (4.6%) | 451,471 (6.5%) |
| Indigenous | 26,811 (0.4%) | 12 (0.2%) | 60 (0.3%) | 6 (0.1%) | 26,889 (0.4%) |
| Mixed | 3,920,721 (56.3%) | 4,467 (73.6%) | 11,033 (58.0%) | 5005 (59.6%) | 3,941,226 (56.4%) |
| Asian | 36,319 (0.5%) | 9 (0.1%) | 103 (0.5%) | 28 (0.3%) | 36,459 (0.5%) |
| Missing | 277,061 (4.0%) | 533 (8.8%) | 932 (4.9%) | 350 (4.2%) | 278,876 (4.0%) |
| **Previous pregnancies** | | | | | |
| None | 4,191,614 (60.2%) | 3574 (58.9%) | 10,560 (55.5%) | 4764 (56.7%) | 4,210,512 (60.2%) |
| ≥1 | 2,768,297 (39.8%) | 2492 (41.1%) | 8462 (44.5%) | 3632 (43.3%) | 2,782,883 (39.8%) |
| **Previous foetal loss** | 1,285,999 (18.5%) | 1275 (21.0%) | 3363 (17.7%) | 1651 (19.7%) | 1,292,288 (18.5%) |
| Missing | 383,632 (5.5%) | 553 (9.1%) | 958 (5.0%) | 420 (5.0%) | 385,563 (5.5%) |
| **Live-born characteristics** | | | | | |
| **Preterm birth (<37 weeks)** | 692,170 (9.9%) | 578 (9.5%) | 1,722 (9.1%) | 821 (9.8%) | 695,291 (9.9%) |
| **Low birth weight (<2,500 g)** | 522,125 (7.5%) | 404 (6.7%) | 1,399 (7.4%) | 654 (7.8%) | 524,582 (7.5%) |
| **Congenital anomaly** | 66,004 (0.9%) | 55 (0.9%) | 192 (1.0%) | 145 (1.7%) | 66,396 (0.9%) |
| **Apgar score 5', median (IQR)** | 9.00 (9.00–10.00) | 9.00 (9.00, 10.00) | 9.00 (9.00, 10.00) | 9.00 (9.00, 10.00) | 9.00 (9.00, 10.00) |
| Missing | 79,419 | 88 | 218 | 51 | 79,776 |
| **Low Apgar score 5' ( < 7)** | 61,905 (0.9%) | 69 (1.2%) | 183 (1.0%) | 84 (1.0%) | 62,241 (0.9%) |
| Missing | 79,419 | 88 | 218 | 51 | 79,776 |
| **Weight for gestational age** | | | | | |
| AGA | 5,387,318 (77.4%) | 4,559 (75.2%) | 14,991 (78.8%) | 6333 (75.4%) | 5,413,201 (77.4%) |
| SGA | 489,981 (7.0%) | 456 (7.5%) | 1446 (7.6%) | 673 (8.0%) | 492,556 (7.0%) |
| LGA | 1,082,612 (15.6%) | 1051 (17.3%) | 2585 (13.6%) | 1390 (16.6%) | 1,087,638 (15.6%) |
| **Neonatal Death** | 43,469 (0.6%) | 53 (0.9%) | 118 (0.6%) | 51 (0.6%) | 43,979 (0.6%) |
| **Pregnancy Arbovirus Symptomatic Infection** | | | | | |
| **Gestational age (days) on arbovirus symptom onset** | – | 158 (97, 213) | 134 (70, 199) | 163 (106, 213) | 147 (84, 205) |
| **Diagnosis criteria** | | | | | |
| Laboratory | – | 3144 (51.8%) | 6,316 (33.2%) | 3,659 (43.6%) | 13,119 (39.2%) |
| Clinical epidemiology | – | 2922 (48.2%) | 12,706 (66.8%) | 4737 (56.4%) | 20,365 (60.8%) |
| **Pre-conception Arbovirus Symptomatic Infection** | | | | | |
| **Any arbovirus infection before pregnancy** | 52,579 (0.8%) | 66 (1.1%) | 226 (1.2%) | 58 (0.7%) | 52,929 (0.8%) |
| **Pre-conception chikungunya** | 4125 (0.1%) | 10 (0.2%) | 6 ( < 0.1%) | <5 ( < 0.1%) | 4143 (0.1%) |
| **Days from pre-conception chikungunya to conception date** | 419 (294, 559) | 326 (280, 450) | 406 (238, 631) | 598 (567, 628) | 419 (294, 559) |
| **Pre-conception dengue** | 45,901 (0.7%) | 49 (0.8%) | 215 (1.1%) | 53 (0.6%) | 46,218 (0.7%) |
| **Days from pre-conception dengue to conception date** | 468 (327, 604) | 447 (240, 576) | 403 (279, 593) | 409 (240, 569) | 468 (326, 604) |
| **pre-conception Zika** | 3082 ( < 0.1%) | 9 (0.1%) | 7 ( < 0.1%) | <5 ( < 0.1%) | 3,101 (0.0%) |
| **Days from pre-conception Zika to conception date** | 445 (313, 571) | 396 (251, 599) | 191 (186, 368) | 229 (223, 274) | 444 (312, 571) |

The raw percentages on the outcomes dependent of gestational age, such as preterm birth and low birth weight do not exhibit direct relationship with the estimated hazard ratio due to the infections being treated as a time-varying variable.

*IQR* interquartile range, *AGA* adequate for gestational age, *SGA* small for gestational age, *LGA* large for gestational age.

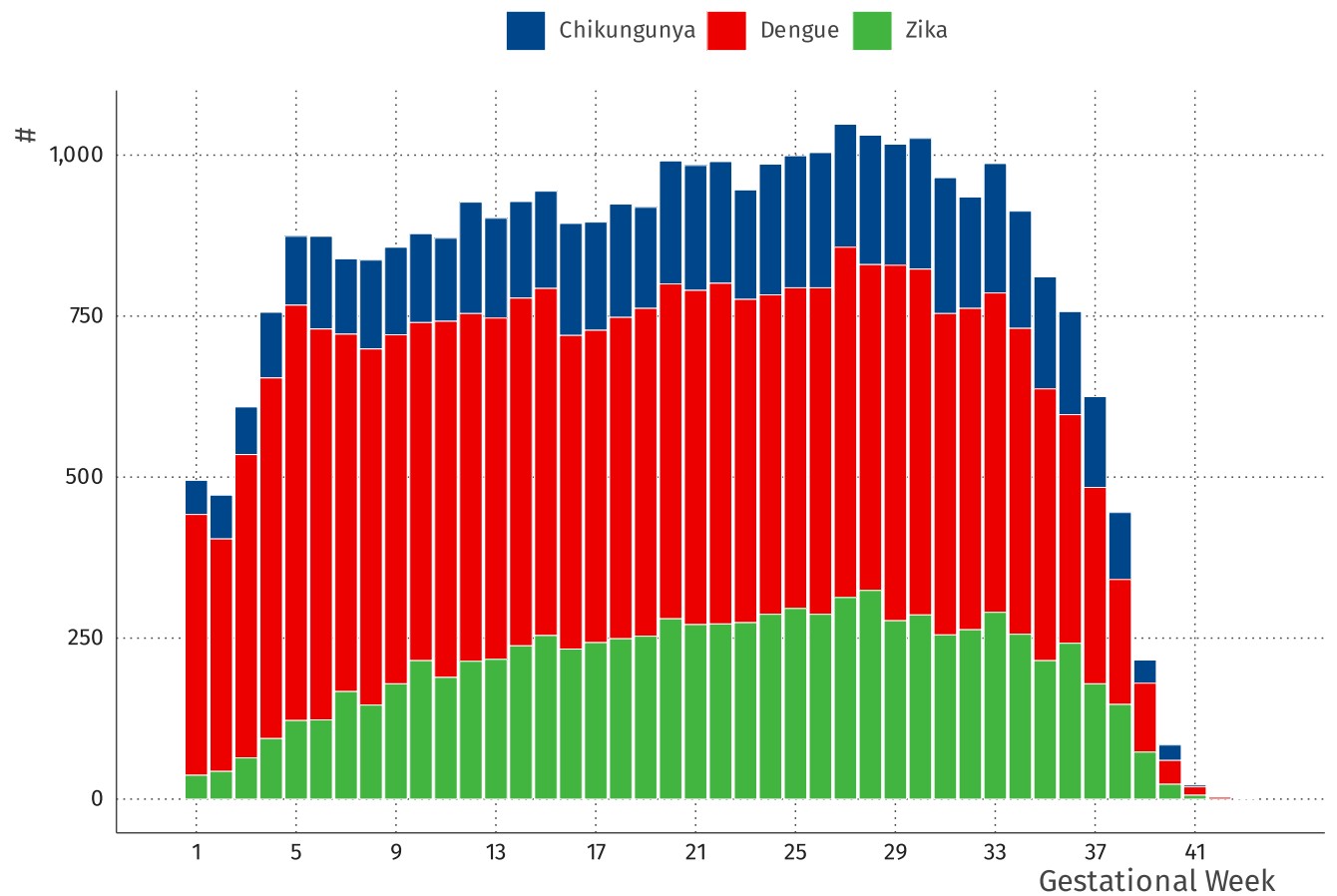

**Fig. 2 | Symptom onset by gestational week.** Distribution of pregnant women with arbovirus infection by gestational week, stratified by virus. Blue represents the number of pregnant women with symptomatic chikungunya infection; red represents the number of pregnant women with symptomatic dengue infection; green represents the number of pregnant women with symptomatic Zika infection.

age distribution was similar among the four comparison groups. Live-born infants exposed to chikungunya were more likely to be from the Northeast region, with a higher proportion of mothers identifying as mixed race, compared to those exposed to dengue, Zika, and unexposed live births. The median gestational age at symptom onset for the three infections was similar, overall 147 days (IQR: 84 to 205) (Table 1, Supplementary Table 1, Fig. 2). A total of 43,979 (0.6%) neonatal deaths occurred in the study, among them 30287 (68.9%) were from preterm births and 8154 (18.5%) were from live births with a detected anomaly.

### Chikungunya
Symptomatic maternal chikungunya was associated with an increased risk of preterm birth, low Apgar score 5', and neonatal death (Fig. 3, Supplementary Table 3). The risk difference for neonatal death, comparing chikungunya-exposed to those unexposed to any infection, was 31.40 (95% CI 6.09–55.92) per 10,000 chikungunya-exposed (Supplementary Table 7). When analysed by trimester, statistically significant heterogeneity was observed for LBW ($p = 0.01$), SGA ($p = 0.02$), congenital anomalies ($p < 0.01$), and low Apgar 5' ($p = 0.04$). (Supplementary Table 2) Infections in the second and third trimesters were associated with higher risks of LBW, low 5-minute Apgar scores, and neonatal death, though the latter showed no statistically significant heterogeneity. Third-trimester infections also increased the risk of congenital anomalies. The analysis restricted to laboratory-confirmed cases showed the same patterns, but with wider confidence intervals (Supplementary Tables 4, 6, 8).

### Dengue
Symptomatic maternal dengue was associated with preterm birth, LBW, congenital anomalies, and low Apgar score 5' (Fig. 3,

Supplementary Table 3). When analysed by trimester, significant heterogeneity was observed for preterm birth ($p = 0.02$), LBW ($p = 0.04$), SGA ($p = 0.02$), and low Apgar 5' ($p = 0.04$). (Supplementary Table 2) Third-trimester infections were associated with a higher risk of preterm birth and low Apgar scores at 5 minutes, while infections in both the second and third trimesters increased the risk of low birth weight. (Fig. 4, Supplementary Table 5) The same patterns were seen in cases with laboratory-confirmed dengue, with slightly higher point estimates and wider confidence intervals. Additionally, laboratory-confirmed cases showed an overall increased risk of neonatal death (Supplementary Tables 4, 6, 8).

### Zika
Symptomatic maternal Zika was associated with all the adverse birth outcomes that were considered. (Fig. 2 and Supplementary Table 3) Live births from pregnancies exposed to Zika had more than twice the risk of congenital anomalies compared to those unexposed to any infection (Fig. 3, Supplementary Table 2). In the analysis by trimester, heterogeneity was observed for congenital anomalies ($p < 0.001$) and neonatal death ($p = 0.02$). (Supplementary Table 2) The risk of congenital anomalies was higher in both the first and third trimesters, while the risk of neonatal death was increased only in the first trimester infections (Fig. 4, Supplementary Table 4). The analysis restricted to laboratory-confirmed cases showed similar results but with wider confidence intervals (Supplementary Tables 4, 6, 8).

### Sensitivity analyses
Our negative control exposure analysis, using pre-conception infection, suggested minimal residual confounding, as the estimates were

| Outcomes | Events | | Estimate (95% CI) |
|---|---|---|---|
| **Preterm** | | | |
| Unexposed | 692,170 | | |
| Chikungunya | 578 | | 1.11 (1.02-1.21) |
| Dengue | 1,722 | | 1.07 (1.02-1.12) |
| Zika | 821 | | 1.25 (1.16-1.34) |
| NCE | 5,192 | | 1.04 (1.01-1.07) |
| **Low birth weight** | | | |
| Unexposed | 522,125 | | |
| Chikungunya | 404 | | 1.10 (1.00-1.22) |
| Dengue | 1,399 | | 1.10 (1.04-1.15) |
| Zika | 654 | | 1.27 (1.18-1.37) |
| NCE | 3,934 | | 1.03 (1.00-1.06) |
| **Small for gestational age** | | | |
| Unexposed | 489,981 | | |
| Chikungunya | 456 | | 1.03 (0.94-1.13) |
| Dengue | 1,446 | | 1.00 (0.95-1.05) |
| Zika | 673 | | 1.20 (1.11-1.29) |
| NCE | 3,776 | | 1.00 (0.96-1.03) |
| **Large for gestational age** | | | |
| Unexposed | 1,082,612 | | |
| Chikungunya | 1,051 | | 1.02 (0.96-1.09) |
| Dengue | 2,585 | | 0.95 (0.91-0.99) |
| Zika | 1,390 | | 1.12 (1.06-1.18) |
| NCE | 8,073 | | 1.02 (1.00-1.04) |
| **Congenital anomalies** | | | |
| Unexposed | 66,013 | | |
| Chikungunya | 55 | | 1.06 (0.81-1.38) |
| Dengue | 189 | | 1.19 (1.03-1.37) |
| Zika | 139 | | 2.36 (2.01-2.79) |
| NCE | 496 | | 1.07 (0.98-1.16) |
| **Low Apgar 5'** | | | |
| Unexposed | 61,905 | | |
| Chikungunya | 69 | | 1.44 (1.14-1.82) |
| Dengue | 183 | | 1.26 (1.09-1.45) |
| Zika | 84 | | 1.31 (1.06-1.63) |
| NCE | 451 | | 1.03 (0.94-1.13) |
| **Neonatal Death** | | | |
| Unexposed | 43,756 | | |
| Chikungunya | 53 | | 1.50 (1.15-1.96) |
| Dengue | 118 | | 1.10 (0.92-1.32) |
| Zika | 51 | | 1.09 (0.83-1.44) |
| NCE | 278 | | 0.94 (0.83-1.05) |

0.5   1.0   3.0

**Fig. 3 | Overall effect estimates.** Estimated adjusted hazard ratios (birth outcomes) and adjusted risk ratios (neonatal death), comparing groups exposed and unexposed to arbovirus infection during pregnancy by outcome. Error bars represent the 95% confidence interval. NCE negative control exposure.

generally closer to the null than the results for the main analyses (especially by trimester), with small associations detected for preterm birth, LBW, and LGA (Fig. 3). The analysis using the missing indicator yielded consistent results with those of multiple imputations for all evaluated outcomes (Supplementary Tables 3–7). Similarly, the analysis using complete-case data yielded similar point estimates for all outcomes, and slightly wider confidence intervals (Supplementary Table 9).

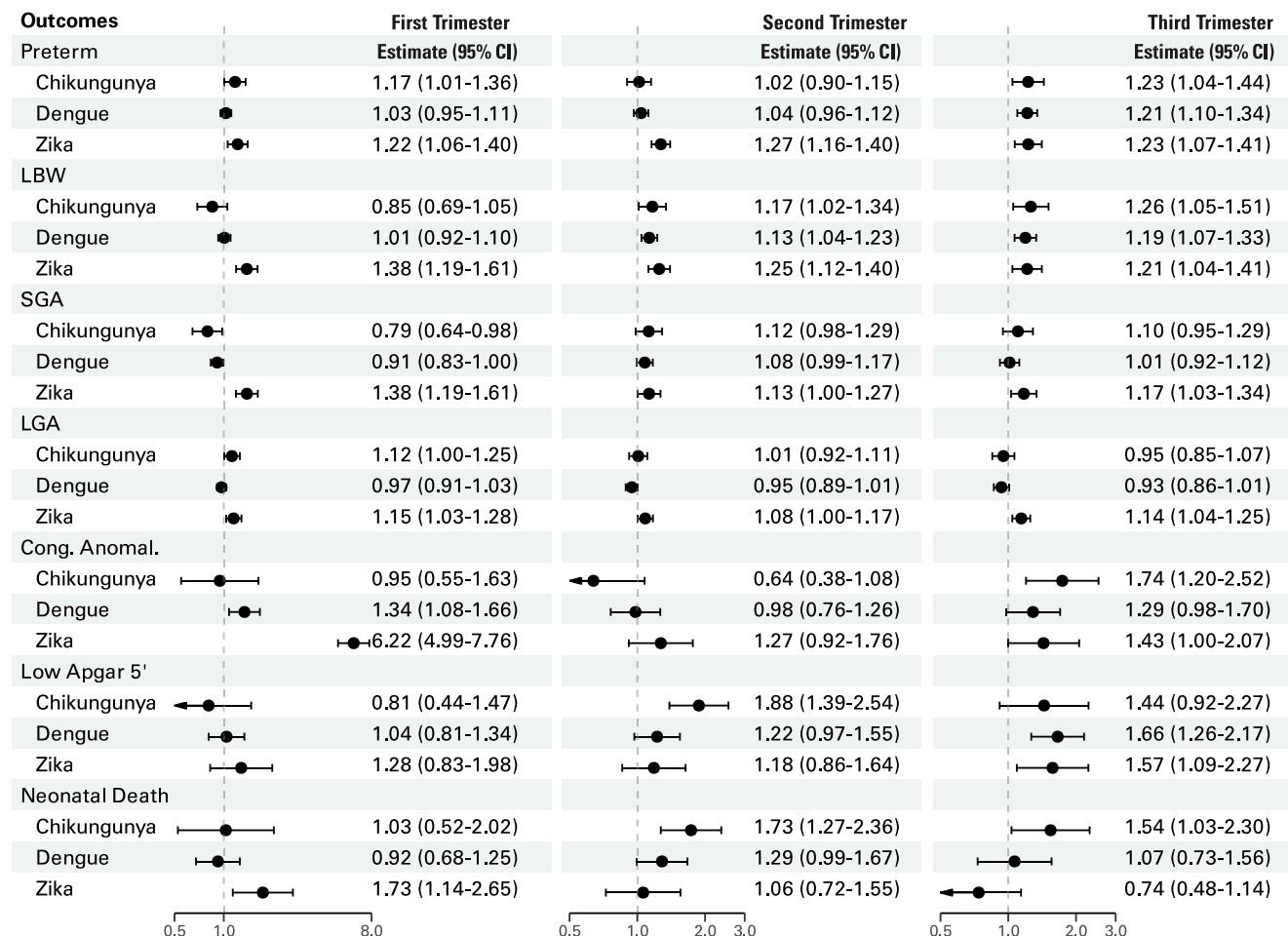

**Fig. 4 | Trimester effect estimates.** Estimated adjusted hazard ratios (birth outcomes) and adjusted risk ratios (neonatal death), comparing groups exposed and unexposed to arbovirus infection during pregnancy by trimester of disease and outcome. The number of pregnant women with symptomatic chikungunya, dengue and Zika infection by trimester is displayed in Supplementary Table 5. Error bars represent the 95% confidence interval.

In the analysis to assess immortal time bias, i.e. not using time-varying exposure, we observed the expected attenuation of results for all outcomes, notably for preterm and LBW, with a change in the direction of the association (Supplementary Fig. 2).

## Discussion

In this nationwide study, we found that maternal chikungunya, dengue, and Zika during pregnancy were associated with increased risks of adverse birth outcomes and neonatal death. Maternal Zika had the strongest associations with most outcomes and was the only virus linked to increased risks of both small and large for gestational age (SGA and LGA) births. The patterns of association varied by trimester of infection: Zika poses the highest risk of various outcomes, acting mainly in the first trimester, including an 80% increased risk of neonatal death and 510% in congenital anomalies; chikungunya and dengue are most impactful in the second and third trimesters, respectively.

Our findings are consistent with previous research on the adverse consequences of maternal Zika infection, particularly during the first trimester[4–6]. Zika is known for its neurotropism and ability to cross the placental barrier, disrupting neurogenesis and development, which can result in congenital malformations such as microcephaly[12]. Our study shows that Zika imposes an additional burden, including increased risks of preterm birth and small for gestational age across infection in all trimesters. Regarding chikungunya, previous studies with small samples (fewer than 200 patients) indicated increased risks

of preterm birth and neonatal complications[8,10,20,22]. However, the largest previous study, which evaluated 1400 pregnancies from the La Réunion outbreak (658 exposed to chikungunya), found no statistically significant differences in adverse birth outcomes[23]. In contrast, our study found that maternal chikungunya was associated with preterm birth, low Apgar 5′ score, and neonatal death. For dengue, there is discordant literature regarding adverse birth outcomes. A meta-analysis from 2022 evaluating dengue during pregnancy, without considering the trimester of infection, found increased risks of stillbirth and neonatal death, but not LBW and preterm, with most of the studies being of moderate quality[13].

We found an increased risk of congenital anomalies following infections during the third trimester for all three infections. This is unusual, as most congenital anomalies occur due to exposures in the first trimester. However, recent evidence highlights different mechanisms, such as acting on cells at the maternal-foetal interface and the impact of pregnancy-related changes on the maternal immune response through which infections may cause congenital anomalies[26]. This means that infections during and of the three trimesters can cause congenital anomalies. This is supported by previous research from Brazil showing abnormal findings in up to 29%(10/36) of Zika infections occurring in the third trimester, including abnormal neurologic exam findings(hypertonicity), polydactyly, microcephaly, and others[27]. For chikungunya, previous studies reported microcephaly, cleft lip, polydactyly, club foot, omphalocele, etc., after third-trimester infections[21,22,28]. We found no reports of congenital anomalies after

third-trimester infections of dengue. We hypothesise that this association was not previously reported due to the large sample size needed to detect a statistically significant result in such a rare event (1% of cases). However, this finding warrants further investigation.

Following chikungunya and Zika, we found an increased risk of LGA, which may be mediated by gestational diabetes. We hypothesise that maternal exposure to chikungunya induces elevated blood glucose levels, resulting in increased glucose transfer to the foetus and excessive growth. Chikungunya infection has been extensively linked to glucose metabolism disorders[29] and also increases the risk of mortality due to diabetes[30]. However, a better understanding of the pathophysiological mechanisms underlying foetal congenital disorders and arbovirus infections is needed.

Our study's key strengths include that we analysed infection as a time-varying exposure to address immortal time bias[24,25], demonstrating that this bias would have occurred for most outcomes if we had not done so. We also stratified the analyses by maternal place of residence and restricted the unexposed group from municipalities with more than 5 cases of arbovirus infections; live births from exposed women to one of the viruses were all compared to the same reference group, i.e. live births from women without evidence of any evidence of infection of the three viruses. Making our results comparable across all three infections.

However, our study has several limitations. First, it relied on registry data with limited access to relevant variables, potentially leading to residual confounding. For example, we lack data on maternal comorbidities, which could increase the risk of infection and adverse pregnancy outcomes. To address this, we used a negative control exposure (infection before pregnancy) to test for spurious associations, and in this analysis, we detected residual confounding. Second, our results could be affected by misclassification since only 30–50% of cases had laboratory confirmation. Nevertheless, because Brazil regularly experiences chikungunya, dengue, and Zika outbreaks, clear guidelines exist for diagnosing these diseases, all requiring mandatory reporting. Notably, our analysis of only laboratory-confirmed cases showed similar increases in adverse birth outcomes, including neonatal death. Third, our study included only pregnant women presenting with symptomatic arboviral infections, as comprehensive testing for arboviruses is not routinely conducted for all pregnant women. Consequently, mild cases that did not require a healthcare visit or asymptomatic cases were not classified as exposed. This may attenuate our findings compared to true effects since women with few or no symptoms would be treated as unexposed. Fourth, in Brazil, gestational age for most live births is estimated by the mother's last menstrual period rather than by ultrasound. Because LMP-based estimates are less precise, this can lead to misclassification of key outcomes—particularly preterm birth and small-for-gestational-age status[31]. Fifth, we lacked information on other diseases that could impact pregnancy, such as cytomegalovirus; however, we excluded any suspected case of arbovirus without confirmation, likely related to other febrile diseases. Sixth, we lacked information regarding previous pregnancies from the same mother, preventing us from adjusting for genetic predisposition to congenital anomalies. Seventh, we did not have data on stillbirths, miscarriages, or abortions, which hindered our ability to evaluate the potential increased risks of these infections causing stillbirths. Additionally, the absence of data on stillbirths or miscarriages can result in "live birth bias", i.e. collider-stratification bias, potentially underestimating the adverse consequences of arbovirus infections on birth and neonatal outcomes if these infections increase the risk of stillbirths. Eighth, we excluded multiple births (2% of the initial sample) because we are unable to identify which live births were from the same pregnancy. Since multiple births are linked to a higher risk of adverse outcomes, this could bias our analysis if most were from unexposed women. However, because our exposure (arbovirus infection) occurs after conception, it is very unlikely that

there is any association between the exposure and the occurrence of multiple births. In this scenario, with rare multiple births and no association between multiple births and the exposure, estimates from approaches that restrict on singletons will yield similar results to those evaluated at the infant level[32]. Lastly, as part of our aim, we wanted to explore differences by trimester of infection. However, even in this large study, we lacked the power to analyse most exposure-outcome associations by trimester.

Our findings of increased risks for key perinatal outcomes have important implications, especially as climate change is likely to increase the mosquito vectors for these viruses globally, including in areas previously unaffected by arboviruses, such as Europe[33]. This study provides robust evidence of the negative consequences of arbovirus infections during pregnancy, including critical windows by trimester. Our findings are relevant to the Sustainable Development Goals 2030 Agenda and the WHO Global Arbovirus Initiative and their concerns with reducing the adverse impacts of neglected tropical diseases[34,35]. They highlight the importance of ongoing research and developing effective antivirals and immunotherapies against chikungunya, dengue, and Zika. Effective vector control, public health surveillance, and educational campaigns to prevent mosquito reservoirs are critical strategies to mitigate these risks. Developing vaccines against these viruses and ensuring equitable access to them in countries with recurring outbreaks as soon as they are approved is crucial for protecting maternal and foetal health worldwide.

## Methods

### Ethical approval
The Federal University of Bahia's Institute of Public Health Ethics Committee provided ethical approval for the study (CAAE registration number: 18022319.4.0000.5030). Informed consent was waived because the data were deidentified and analysed under strict security procedures, in accordance with the General Data Protection Law (13,709/2018), Article 7, Item IV.

### Study population and data source
We conducted a population-based cohort study of live births in Brazil, among pregnant women aged 12–49 years, with estimated conception dates between January 1, 2015, and March 1, 2019. By restricting follow-up to live births observed through February 2020, we aimed to mitigate the possibility of misclassifying the exposure (i.e. classify a COVID-19 case as arbovirus infection).

The source population was drawn from the national live birth registry, the Live Birth Information System (Sistema de Informação de Nascidos Vivos, SINASC). The SINASC provided information about the mother, including maternal age at birth, education level, marital status and ethnicity, the pregnancy (prenatal appointments, parity, previous loss, length of gestation) and the newborn (birthweight, sex, Apgar score, presence of congenital anomalies). The congenital anomalies recorded in SINASC are those detected at birth and documented according to the International Classification of Diseases–10.

Data on symptomatic chikungunya, dengue and Zika records were obtained from the Notifiable Diseases Information System (Sistema de Informação de Agravos de Notificação; SINAN), which included information on the notified infections: date of symptom onset, diagnosis and criteria used for diagnosis. In Brazil, all suspected cases of chikungunya, dengue and Zika are recorded, and laboratory confirmation is required until a predefined incidence threshold is reached. Once this threshold is reached, diagnosis can be based on clinical-epidemiological criteria, considering typical symptoms occurring in the same area and timeframe as other confirmed cases and validation by epidemiological surveillance[36]. Detailed information on chikungunya, dengue and Zika case definitions can be found in our previously published protocol[36].

Data on the cause and date of death of the live births were obtained from the Mortality Information System (Sistema de Informação sobre Mortalidade, SIM).

We excluded: (i) live births before 22 weeks or after 44 weeks of gestation or with a birthweight below 500 g; (ii) non-singleton live births; (iii) live births with missing data on the municipality of residence; (iv) live births from women with a suspected case of chikungunya, dengue or Zika in the six months before the conception date up to 6 days after conception date; (v) for the exposed women, we excluded live births from women with more than one confirmed arbovirus infection during pregnancy; and (vi) live births from women notified as suspected arbovirus infections ruled out after clinical and epidemiologic investigation, as those records are likely to be cases of other febrile diseases. Finally, we restricted the analysis to municipalities with at least five confirmed cases of at least one of the three arboviruses to prevent sparse data bias[37].

### Linkage process
The SINASC records were linked separately to SINAN chikungunya, SINAN dengue, SINAN Zika, and SIM records using variables such as maternal name, date of birth or age, and place of residency as matching criteria. This linkage was performed using CIDACS-Record Linkage, which combines indexing and searching algorithms to identify records from the SINASC that closely match each record in the remaining datasets. It proceeds with pairwise comparisons of candidate linking records[38]. The accuracy of each linkage was assessed through manual verification of a randomly selected sample of records, evaluating sensitivity and specificity indexes via receiver operating characteristic curves. More details about the linkage procedures are provided in previously published articles[38,39], and the performance metrics are presented in Supplementary Fig. 3.

### Exposures and outcomes
Live births from SINASC that were linked with SINAN records indicating that the mother was reported and confirmed as an arboviruses case during pregnancy (occurring between seven or more days after the estimated date of conception and the date of delivery) were considered exposed. The date of conception was estimated using the birth date minus the gestational age at birth as recorded in SINASC dataset. We used seven days after conception to reduce potential misclassification of pre-pregnancy infection due to the time of incubation of those viruses. Pregnancies not linked to a record of suspected chikungunya, dengue or Zika during pregnancy were considered unexposed. The trimester of infection during pregnancy was classified as the first trimester (pregnancy day 7–97 days), the second trimester (98–195 days), or the third trimester (196 days to birth). Supplementary Fig. 1 shows a representation of the time-varying classification of infection status.

We defined preterm birth as delivery before 37 completed weeks of gestation. The risk window for preterm birth was from 22 to 37 weeks of gestation (pregnancy day 258). Large for gestational age (LGA) and small for gestational age (SGA) at birth were defined as above the 90th and below the 10th centile of the sex-specific birthweight-for-gestational-age distribution based on the Intergrowth reference charts, respectively[40]. Low birth weight (LBW) was defined as infants weighting <2.5 kg. Low Apgar was defined as a score of less than 7 (maximum of 10) 5 minutes after birth. We also identified the occurrence of congenital anomalies recorded at SINASC after birth (International Classification of Diseases, 10th revision codes Q00-Q99). The risk window for LBW, LGA, SGA, congenital anomaly and Apgar 5' < 7, it was 22 weeks to the end of pregnancy. Neonatal death was defined as death occurring within 27 days of birth.

### Covariables
We considered the following to be a priori confounders based on their known or plausible causal effects on infection risk and/or adverse pregnancy outcomes (disjunctive cause criterion): mother's age (<20, 20-34, ≥ 35), education (0–3, 4–7, 8–11 or ≥12 years), previous pregnancies (0 or ≥1), number of prenatal appointments (0, 1–3, 4–6, ≥7), previous stillbirth, marital status (married/stable union or single), municipality of residence, race (white, black, mixed race, Asian and Indigenous) and year of conception. Records with missing data on the municipality of residence or on sex of the live birth were excluded (6,322 - <0.01%); missing data on any of the other confounders (695,829 −10%) was addressed using multiple imputation.

### Statistical analyses
For the birth outcomes, we used a Cox model to estimate the hazard ratio (HR) with a 95% confidence interval (CI), stratified by municipality (i.e. area of residence). Infection was considered a time-varying exposure, meaning that pregnant women were classified as unexposed before infection and exposed after infection. Robust sandwich variance estimation was used to account for statistical dependence across repeated observations because of changes in exposure status. Gestational age in days was used as the timescale. In additional analyses, we explored whether associations differed by trimester using the same analysis approach described above. Statistical evidence for a difference between trimesters was assessed using Cochran's Q test. We assessed the proportional hazards assumption using a test based on Schönfeld residuals[41].

For neonatal death, we used Poisson regression with clustered (municipality) standard errors to derive risk ratios (RR) and risk difference (RD); the 95% CI for RD was estimated using the delta method.

Multiple imputation was conducted using chained equations using a fully conditional specification to generate six imputed datasets with five iterations per dataset, using random forest as the underlying model.

### Sensitivity analyses
We repeated the analyses using laboratory-confirmed infection only to explore potential exposure misclassification bias due to including participants without laboratory diagnosis.

We conducted a negative control exposure analysis[42], using maternal arbovirus infection that occurred between 6 and 24 months before pregnancy, under the assumption that maternal arbovirus infection before pregnancy is likely to be influenced by many of the same unmeasured potential confounders as maternal infection during pregnancy, but is not expected to have any direct causal influence on birth and neonatal outcomes.

We also assessed the consistency of the results regarding missing data. In additional analyses, we employed the missing indicator method and complete-case analysis. Each analysis considers a different type of underlying mechanism for the missing data (at random, completely at random or not at random)[43].

Finally, to assess how immortal time bias could impact the results, we classified the live-born as exposed since conception if their mother had an infection at any point during the pregnancy, i.e. not defining arbovirus infection as a time-varying exposure. We derived RRs using the Poisson regression with clustered standard errors per municipality of maternal residence.

All data processing and analyses were done using R (version 4.1.1) and the tidyverse, fixest, mice, marginaleffects and survival packages.

### Reporting summary
Further information on research design is available in the Nature Portfolio Reporting Summary linked to this article.

## Data availability

The relevant data are available in the manuscript and the Supplementary Information. Raw data are available upon reasonable request to the Centro de Integração de Dados e Conhecimentos para a Saúde (CIDACS). Any person who wishes to receive authorisation must: (1) be affiliated to CIDACS or be accepted as collaborators; (2) present a detailed research project together with approval by an appropriate Brazilian institutional research ethics committee; (3) provide a clear data plan restricted to the objectives of the proposed study and a summary of the analyses plan intended to guide the linkage and data extraction of the relevant set of records and variables; (4) sign terms of responsibility regarding the access and use of data; and (5) perform the analyses of datasets provided using the CIDACS data environment, a safe and secure infrastructure that provides remote access to de-identified or anonymised datasets and analysis tools. For more information: https://cidacs.bahia.fiocruz.br/.

## Code availability

The modelling in this paper used R v.4.1.1 and the tidyverse v.1.3.2, survival v.3.7-0, fixest v.0.11.2, mice v.3.17, and marginaleffects 0.18.0 R packages, all of which are freely available.

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

## Acknowledgements

This study was financed in part by the Brazilian National Research Council (CNPq). M.L.B. and VSB are Brazilian National Research Council research fellows. E.S.P. acknowledges funding from the Wellcome Trust (225925/Z/22/Z). T.C.-S. acknowledges funding from the Royal Society (NIF\R1\231435). D.A.L.'s contribution is supported by the UK Medical Research Council (MC_UU_00032/05).

## Author contributions

T.C.-S. and E.S.P. conceptualised the study. T.C.-S. wrote the first draft of the manuscript. N.P. and D.A.L. supervised the data analysis. T.C.-S. conducted the formal analysis. V.S.B., L.C.R., E.S.P., N.P., D.A.L. and M.L.B. critically review the manuscript. M.G.T., M.C.N.C., and L.C. contributed to the writing and editing of the manuscript. T.C.-S. decided to submit the manuscript for publication. T.C.-S. and E.S.P. had access to the raw data in the study and accessed and verified the data.

## Competing interests

The authors declare no competing interests.
