## [Peer review file · Nature Communications]

Perinatal outcomes of symptomatic chikungunya, dengue and Zika infection during pregnancy in Brazil: a registry-based cohort study

Corresponding Author: Dr Thiago Cerqueira-Silva

Version 0:

Reviewer comments:

Reviewer #1

(Remarks to the Author)

Studies on the effects of maternal infections by urban arboviruses on pregnancy outcomes and perinatal mortality have yielded conflicting results. In order to contribute to this discussion, the authors used the large national health information systems from Brazil, comprising a cohort of almost 7 million live births. The association between these infections and most of the outcomes analyzed was demonstrated, particularly for infections by the Zika virus. When I started reading the study, my concern was related to my observation, from my own experience, that only about 1/3 of the reported cases of dengue fever in Brazil, on average, have laboratory confirmation, and about 50% of the cases of Zika and chikungunya. However, the authors performed a sensitivity analysis, including only cases with laboratory confirmation. Therefore, I believe that my main concern was adequately addressed by the authors.

Reviewer #2

(Remarks to the Author)

The authors examined the risk of perinatal outcomes among pregnant women with symptomatic chikungunya, dengue, and Zika virus using a population-based cohort in Brazil and reported that symptomatic maternal chikungunya to be associated with an increased risk of preterm birth, low Apgar score 5' and neonatal death; symptomatic maternal dengue to be associated with preterm birth, low birth weight, congenital anomalies, and low Apgar score 5' and symptomatic maternal Zika associated with all adverse birth outcomes, particularly congenital anomalies.

The study is not novel per se except that it examined all of these exposures in a single cohort. In general, the writing is good and all sections are good except perhaps the sensitivity analysis section, description of the exposures, and Table 1 that needs clarification or modification.

SPECIFIC COMMENTS

METHODS

Exposure was infection 7 days or more after conception date and women were considered unexposed if the pregnancy was not linked to a case of chikungunya, dengue or Zika during pregnancy. However, it is unclear if the unexposed group was followed 7 days or more after conception date or the entire pregnancy (lines 148-149). The time-period should be the same for both exposed and unexposed.

Please describe what source was used to classify congenital anomalies because not all congenital anomalies are important/major. For example, was it the Eurocat classification or something else?

Please explain why only singleton births were included? This is likely to underestimate the preterm births because a higher proportion of non-singleton births are likely to be born preterm.

Lines 136-139: One can find the details about the linkage in a previous publication. However, it would be helpful to provide brief information on the accuracy of each linkage, sensitivity, and specificity, and AUROC.

Please explain how the exposure (infection status) was assessed and add a variable about gestational age (mean/median) at diagnosis to Table 1 and the text.

The study outcomes included preterm birth, low birthweight, sex-specific birthweight for gestational age, low Apgar score, congenital anomalies, and neonatal death. All of these outcomes except neonatal death are birth outcomes. Therefore, it would be more appropriate to change perinatal outcomes to birth and neonatal outcomes.

Lines 164- 173: Though the authors have described that these are confounders some of the variables such as previous pregnancies, number of prenatal appointments, previous stillbirth, and year of conception are not confounders but covariates because these are only associated with the outcome and not the exposure.

It would be incorrect to say that the risk window is from 22 weeks onwards especially for congenital anomalies because as commented above the sensitive window for different systems are different and different exposures are likely to have differential impact not only because of the infectiousness of the exposure but also because of the critical period of development for different organs/systems.

There are issues with the 'Sensitivity analyses' section that needs to be clarified or modified.

1. Please could you specify what infection was included for the negative controls. Additionally, one key assumption made in negative control studies is that the relationship between the exposure and outcome and the control and outcome are subject to the same confounding. However, this is not true for the current study because number of prenatal appointments and year of conception are less likely to be confounders of the negative exposure control and the outcomes.

2. Lines 195-196: Please modify this to something more appropriate because neonatal death (one of the study outcomes) is not a perinatal outcome.

3. Using missing indicator method for missing data does not add any new information. Also, missing indicator method to account for missing data is known to be a biased method and not superior to multiple imputation. Please refer to: Donders AR, van der Heijden GJ, Stijnen T, et al. Review: a gentle introduction to imputation of missing values. Instead, a complete case analysis is more informative. It would also be informative to compare the characteristics of women with and without missing data and to provide the extent of missing values.

4. Immortal time bias: It is good that the authors used time-varying exposure to account for immortal time bias. However, the sensitivity analysis related to immortal bias seems incorrect. All the outcomes are at and after birth and after using time-varying exposure immortal time bias is not likely to be a problem in the current study but collider bias could be a problem due to the inclusion of only live births.

RESULTS

Also, it would be informative to have a table with n (%) for the congenital anomalies probably by system (because of the small numbers) to evaluate the critical window of exposure because different systems/organs have different sensitive periods of development and the influence of teratogens is closely related to this.

Please clarify how symptomatic chikungunya, dengue, and zika was defined.

Table 1. Of the entire paper this is perhaps the only section that seems to be of low quality.

For example, it is unclear what the variables, days from chikungunya to conception date indicate. Is this for the current infection or days from previous infection? Similarly, it is unclear what the days for dengue and Zika viruses mean.

Please clarify what n (%) for clinical epidemiology and laboratory mean in Table 1 and why is it necessary in this table? It is unclear why the two variables for Apgar score are separated by the congenital anomaly variable.

Also, it is difficult to comprehend what the first Apgar score means. Probably, it is the median (IQR) but there is no information provided.

It would be informative to provide gestational age at diagnosis (perhaps, the authors have reported it but since the table is poorly formatted it is not easy to understand the information provided).

Please modify this table to align with the results and also remove unnecessary information. It would be to organise the variables by certain characteristics.

Figure 2: Good and informative figure

Figure 3. Please add the variables adjusted in the models and also change the heading 'Estimate' to meaningful such as Adjusted hazard ratio (95%CI). The negative control does not seem to be robust for preterm birth because there was a significant association the magnitude of which is comparable to chikungunya. This may suggest that the use of a common negative control for all outcomes is not robust.

Please add information on how many neonatal deaths were among preterm births and those who had congenital anomalies. This could be added to the text and not to the table.

Supplementary Table 9. It is not recommended to report numbers less than 5 or 10 in tables especially for rare diseases due to confidentiality reasons and statistical disclosure. Therefore, please combine categories appropriately or report only the percentages.

Supplementary Figure 2 does not provide any useful information. Therefore, it can be deleted.

DISCUSSION

Please note: Neonatal death is not a perinatal outcome.

A study limitation is collider bias because of the inclusion of only live births.

Because stillbirths and pregnancy losses were not included the association for congenital anomalies is likely to be underestimated because a high proportion of pregnancy losses are congenital anomalies.

MINOR COMMENT:

The use of the word, effects, suggests causality which cannot be inferred from most observational studies. Therefore, please change all instances of this to non-causal language.

Version 1:

Reviewer comments:

Reviewer #2

(Remarks to the Author)

Thank you for responding to previous comments and revising the paper accordingly. I have the following comments related to responses to the previous comments.

PREVIOUS COMMENT:

1. Please explain why only singleton births were included? This is likely to underestimate the preterm births because a higher proportion of non-singleton births are likely to be born preterm.

AUTHOR'S RESPONSE:

We excluded all non-singleton births to isolate the effect of arboviral exposure from the well-established risks associated with multiple births. Because our analysis focuses on exposures during pregnancy, including multiple births would result in the same maternal exposure being counted more than once—potentially leading to an overestimation of the effect, given that multiple births are associated with all the outcomes under study. To avoid this, we excluded multiple births from the analysis

COMMENT:

This does not seem to be sufficient reason to exclude multiple births because there are methods to account for clustering due to multiple births such as multilevel modelling. However, if the authors insist on using data from singleton births, then please add this as a study limitation with citations because multiple birth is an important for preterm birth and congenital anomalies.

PREVIOUS COMMENT:

2. Using missing indicator method for missing data does not add any new information. Also, missing indicator method to account for missing data is known to be a biased method and not superior to multiple imputation. Please refer to: Donders AR, van der Heijden GJ, Stijnen T, et al. Review: a gentle introduction to imputation of missing values. Instead, a complete case analysis is more informative. It would also be informative to compare the characteristics of women with and without missing data and to provide the extent of missing values.

AUTHOR'S RESPONSE:

We thank the reviewer for the comment. We have now included a complete-case analysis (Supplementary Table 9, which includes the number per group and the number of events) as a sensitivity analysis, showing similar point estimates with wider confidence intervals in some outcomes.

Below we provide the results of the 3 approaches for 4 outcomes:

We also opted to retain the analysis with the missing indicator, as emerging theoretical and empirical evidence have shown that, under certain circumstances, the missing indicator method can provide valid inferences with minimal computational cost.

Please see (Theoretical):

1. Blake HA, Leyrat C, Mansfield KE, Tomlinson LA, Carpenter J, Williamson EJ. Estimating treatment effects with partially observed covariates using outcome regression with missing indicators. *Biometrical Journal* 2020; 62: 428–43.
2. Zhuchkova, Svetlana, and Aleksei Rotmistrov. "How to choose an approach to handling missing categorical data:(un) expected findings from a simulated statistical experiment." *Quality & Quantity* 56.1 (2022): 1-22.
3. Song, Mingyang, et al. "The missing covariate indicator method is nearly valid almost always." *arXiv preprint arXiv:2111.00138* (2021).

(Empirical)

Other articles with very similar results among multiple imputation and the/ use of missing indicator

1. Cahill, Leah E., et al. "Prospective study of breakfast eating and incident coronary heart disease in a cohort of male US health professionals." *Circulation* 128.4 (2013): 337-343.
2. Joosten, Michel M., et al. "Associations between conventional cardiovascular risk factors and risk of peripheral artery

disease in men." *Jama* 308.16 (2012): 1660-1667.

3. Matthews, Anthony A., et al. "Prospective benchmarking of an observational analysis in the SWEDEHEART registry against the REDUCE-AMI randomized trial." *European Journal of Epidemiology* 39.4 (2024): 349-361.

COMMENT:

Thank you for the information provided regarding the use of the indicator method. However, the purpose of the indicator method analysis is still unclear. In this study, analysis based on multiple imputation is the primary analysis. Therefore, one may expect to compare the results from these analyses with a complete case analysis to assess if missing data could influence the results. However, it is unclear why it was necessary to compare the results from MI with indicator method. Also, could you please describe the 'certain circumstances' (missing indicator method can provide valid inferences with minimal computational cost), with respect to this study.

I may have missed it but there seems to be no response to the suggestion: It would also be informative to compare the characteristics of women with and without missing data and to provide the extent of missing values.

PREVIOUS COMMENT

3. The negative control does not seem to be robust for preterm birth because there was a significant association the magnitude of which is comparable to chikungunya. This may suggest that the use of a common negative control for all outcomes is not robust

AUTHOR'S RESPONSE

No response found

COMMENT:

This comment seems to be unaddressed.

Reviewer #1 (Remarks to the Author):

Studies on the effects of maternal infections by urban arboviruses on pregnancy outcomes and perinatal mortality have yielded conflicting results. In order to contribute to this discussion, the authors used the large national health information systems from Brazil, comprising a cohort of almost 7 million live births. The association between these infections and most of the outcomes analyzed was demonstrated, particularly for infections by the Zika virus. When I started reading the study, my concern was related to my observation, from my own experience, that only about 1/3 of the reported cases of dengue fever in Brazil, on average, have laboratory confirmation, and about 50% of the cases of Zika and chikungunya. However, the authors performed a sensitivity analysis, including only cases with laboratory confirmation. Therefore, I believe that my main concern was adequately addressed by the authors.

R: We thank the reviewer for the positive comments.

Reviewer #2 (Remarks to the Author):

The authors examined the risk of perinatal outcomes among pregnant women with symptomatic chikungunya, dengue, and Zika virus using a population-based cohort in Brazil and reported that symptomatic maternal chikungunya to be associated with an increased risk of preterm birth, low Apgar score 5' and neonatal death; symptomatic maternal dengue to be associated with preterm birth, low birth weight, congenital anomalies, and low Apgar score 5' and symptomatic maternal Zika associated with all adverse birth outcomes, particularly congenital anomalies.

The study is not novel per se except that it examined all of these exposures in a single cohort. In general, the writing is good and all sections are good except perhaps the sensitivity analysis section, description of the exposures, and Table 1 that needs clarification or modification.

R: We thank the reviewer for their positive evaluation and the additional constructive and helpful comments, which have substantially improved the manuscript.

SPECIFIC COMMENTS

METHODS

Exposure was infection 7 days or more after conception date and women were considered unexposed if the pregnancy was not linked to a case of chikungunya,

dengue or Zika during pregnancy. However, it is unclear if the unexposed group was followed 7 days or more after conception date or the entire pregnancy (lines 148-149). The time-period should be the same for both exposed and unexposed.

R: Both groups were followed in the gestational age timescale. In this case, the follow-up starts at the conception date and ends at the date of birth or 37 weeks (for preterm birth), and the time period was not changed for any group. Infections occurring less than 7 days after the conception date were excluded, as there is a risk of misclassification, since the SINASC dataset only provides gestational age in discrete units of gestational weeks (1-week intervals). We have clarified this information in the methods now:

“We excluded: (i) live births before 22 weeks or after 44 weeks of gestation or with a birthweight below 500g; (ii) non-singleton live births; (iii) live births with missing data on the municipality of residence; (iv) live births from women with a suspected case of chikungunya, dengue or Zika in the six months before the conception date up to 6 days after conception date”

Statistical analysis:

“Gestational age in days was used as the timescale.”

Please describe what source was used to classify congenital anomalies because not all congenital anomalies are important/major. For example, was it the Eurocat classification or something else?

R: Information on congenital anomalies was obtained from the SINASC dataset, where all identified anomalies should be recorded. This system serves as a key source for monitoring and is one of the main tools for the surveillance of congenital anomalies in the country. We used ICD-10 codes Q00–Q99. We have included this information in the methods now.

“We also identified the occurrence of congenital anomalies recorded at SINASC after birth (International Classification of Diseases, 10th revision codes Q00-Q99).”

Please explain why only singleton births were included? This is likely to underestimate the preterm births because a higher proportion of non-singleton births are likely to be born preterm.

R: We excluded all non-singleton births to isolate the effect of arboviral exposure from the well-established risks associated with multiple births. Because our analysis focuses on exposures during pregnancy, including multiple births would result in the same maternal exposure being counted more than once—potentially leading to an overestimation of the effect, given that multiple births are associated with all the outcomes under study. To avoid this, we excluded multiple births from the analysis.

Lines 136-139: One can find the details about the linkage in a previous publication. However, it would be helpful to provide brief information on the accuracy of each linkage, sensitivity, and specificity, and AUROC.

R: We have now included the AUROC and performance metrics of the linkage in the supplementary material (Appendix Methods).

Please explain how the exposure (infection status) was assessed and add a variable about gestational age (mean/median) at diagnosis to Table 1 and the text.

R: Exposure to Zika, dengue, and chikungunya was assessed through record linkage between the cohort and the Information System for Notifiable Diseases (SINAN). SINAN only records symptomatic infections. Both clinically and laboratory-confirmed cases were included, and sensitivity analyses were conducted using only laboratory-confirmed cases.

We have included the requested information now.

“Data on symptomatic chikungunya, dengue and Zika records were obtained from the Notifiable Diseases Information System (*Sistema de Informação de Agravos de Notificação*; SINAN), which included information on the notified infections: date of symptom onset, diagnosis and criteria used for diagnosis.”

“Live births from SINASC that were linked with SINAN records indicating that the mother was reported and confirmed as an arboviruses case during pregnancy (occurring between seven or more days after the estimated date of conception and the date of delivery) were considered exposed”

The study outcomes included preterm birth, low birthweight, sex-specific birthweight for gestational age, low Apgar score, congenital anomalies, and neonatal death. All of

these outcomes except neonatal death are birth outcomes. Therefore, it would be more appropriate to change perinatal outcomes to birth and neonatal outcomes.

R: The requested change has been made through the text.

Lines 164- 173: Though the authors have described that these are confounders some of the variables such as previous pregnancies, number of prenatal appointments, previous stillbirth, and year of conception are not confounders but covariates because these are only associated with the outcome and not the exposure.

R: We agree with the reviewer that some of the variables (previous pregnancies and previous stillbirths) are associated only with the outcome. However, number of prenatal appointments can be regarded as proxy for healthcare access/seeking behaviour, making it associated with exposure and outcome, similarly the year of conception is related to the epidemiological situation of the arbovirus of interest in the paper (chikungunya, dengue and Zika) and also associated with trends in the healthcare quality.

Following the common cause criteria/back door criteria, the previous pregnancy/previous stillbirth wouldn't be defined as confounders. Using the disjunctive cause criterion {1}, both variables are confounders. We have rephrased the sentence in the methods to make clear which criteria we used to define confounders.

“We considered the following to be *a priori* confounders based on their known or plausible causal effects on infection risk and/or adverse pregnancy outcomes (disjunctive cause criterion):”

- 1. VanderWeele, Tyler J. "Principles of confounder selection." *European journal of epidemiology* 34 (2019): 211-219.**

It would be incorrect to say that the risk window is from 22 weeks onwards especially for congenital anomalies because as commented above the sensitive window for different systems are different and different exposures are likely to have differential impact not only because of the infectiousness of the exposure but also because of the critical period of development for different organs/systems.

R: As our analysis is restricted to live-births, any outcome can only be detected in live-births, i.e. babies born with at least 22 weeks. The risk window does not reflect the period during which the pregnant woman can have symptomatic

infection. The infection can occur anytime between 7 days after the conception date and the date of birth.

“The trimester of infection during pregnancy was classified as the first trimester (pregnancy day 7 to 97 days), the second trimester (98 to 195 days), or the third trimester (196 days to birth). Supplementary Figure 1 shows a representation of the time-varying classification of infection status.”

Our definition of risk windows is to make clear the period of assessment of the outcome. This is very important for preterm birth, which can only occur before 37 weeks. While the other outcomes can occur after that, being assessed at birth.

There are issues with the ‘Sensitivity analyses’ section that needs to be clarified or modified.

1. Please could you specify what infection was included for the negative controls. Additionally, one key assumption made in negative control studies is that the relationship between the exposure and outcome and the control and outcome are subject to the same confounding. However, this is not true for the current study because number of prenatal appointments and year of conception are less likely to be confounders of the negative exposure control and the outcomes.

R: We have now changed “maternal infection” to “maternal arbovirus infection”.

“We conducted a negative control exposure analysis,³² using maternal arbovirus infection that occurred between 6 and 24 months before pregnancy, under the assumption that maternal arbovirus infection before pregnancy is likely to be influenced”

The number of prenatal appointments can be seen as a proxy of healthcare access/seeking behaviour while year of conception is directly related to the epidemiological situation of the three diseases, for example, the peak of Zika cases in Brazil occurred in 2016. Similarly, chikungunya outbreaks occur in different states in Brazil, with Ceará reporting two big outbreaks in 2016 and 2017. The negative control exposure should share a similar confounding structure, but not necessarily be equal. {1,2} We consider that these variables provide valuable information in that analysis.

1. Liew, Zeyan, et al. "Use of negative control exposure analysis to evaluate confounding: an example of acetaminophen exposure and attention-deficit/hyperactivity disorder in Nurses' Health Study II." *American journal of epidemiology* 188.4 (2019): 768-775.

2. Lipsitch, Marc, Eric Tchetgen Tchetgen, and Ted Cohen. "Negative controls: a tool for detecting confounding and bias in observational studies." *Epidemiology* 21.3 (2010): 383-388.

2. Lines 195-196: Please modify this to something more appropriate because neonatal death (one of the study outcomes) is not a perinatal outcome.

R: We have now changed it to “birth and neonatal outcomes”

3. Using missing indicator method for missing data does not add any new information. Also, missing indicator method to account for missing data is known to be a biased method and not superior to multiple imputation. Please refer to: Donders AR, van der Heijden GJ, Stijnen T, et al. Review: a gentle introduction to imputation of missing values. Instead, a complete case analysis is more informative. It would also be informative to compare the characteristics of women with and without missing data and to provide the extent of missing values.

R: We thank the reviewer for the comment. We have now included a complete-case analysis (Supplementary Table 9, which includes the number per group and the number of events) as a sensitivity analysis, showing similar point estimates with wider confidence intervals in some outcomes.

Below we provide the results of the 3 approaches for 4 outcomes:

Outcomes	Estimate - 95% CI		
	Complete Case	Missing Indicator	Multiple Imputation
Preterm			
Unexposed			
Chikungunya	1.11 (1.01 to 1.21)	1.11 (1.03 to 1.21)	1.11 (1.02 to 1.21)
Dengue	1.06 (1.01 to 1.12)	1.07 (1.02 to 1.12)	1.07 (1.02 to 1.12)
Zika	1.24 (1.15 to 1.33)	1.25 (1.17 to 1.34)	1.25 (1.16 to 1.34)
Low birth weight			
Unexposed			
Chikungunya	1.14 (1.03 to 1.27)	1.11 (1.01 to 1.22)	1.10 (1.00 to 1.22)

Dengue	1.08 (1.03 to 1.15)	1.10 (1.04 to 1.16)	1.10 (1.04 to 1.15)
Zika	1.27 (1.17 to 1.38)	1.27 (1.18 to 1.38)	1.27 (1.18 to 1.37)

Small for gestational age

Unexposed

Chikungunya	1.09 (0.99 to 1.21)	1.04 (0.95 to 1.14)	1.03 (0.94 to 1.13)
Dengue	0.99 (0.94 to 1.05)	1.00 (0.95 to 1.06)	1.00 (0.95 to 1.05)
Zika	1.20 (1.11 to 1.30)	1.20 (1.11 to 1.29)	1.20 (1.11 to 1.29)

Neonatal Death

Unexposed

Chikungunya	1.59 (1.19 to 2.13)	1.50 (1.16 to 1.94)	1.50 (1.16 to 1.93)
Dengue	1.04 (0.85 to 1.27)	1.10 (0.93 to 1.29)	1.10 (0.93 to 1.28)
Zika	1.03 (0.76 to 1.40)	1.09 (0.85 to 1.40)	1.09 (0.85 to 1.39)

We also opted to retain the analysis with the missing indicator, as emerging theoretical and empirical evidence have shown that, under certain circumstances, the missing indicator method can provide valid inferences with minimal computational cost.

Please see (Theoretical):

1. Blake HA, Leyrat C, Mansfield KE, Tomlinson LA, Carpenter J, Williamson EJ. Estimating treatment effects with partially observed covariates using outcome regression with missing indicators. *Biometrical Journal* 2020; 62: 428–43.
2. Zhuchkova, Svetlana, and Aleksei Rotmistrov. "How to choose an approach to handling missing categorical data:(un) expected findings from a simulated statistical experiment." *Quality & Quantity* 56.1 (2022): 1-22.
3. Song, Mingyang, et al. "The missing covariate indicator method is nearly valid almost always." *arXiv preprint arXiv:2111.00138* (2021).

(Empirical)

Other articles with very similar results among multiple imputation and the/ use of missing indicator

1. Cahill, Leah E., et al. "Prospective study of breakfast eating and incident coronary heart disease in a cohort of male US health professionals." *Circulation* 128.4 (2013): 337-343.
2. Joosten, Michel M., et al. "Associations between conventional cardiovascular risk factors and risk of peripheral artery disease in men." *Jama* 308.16 (2012): 1660-1667.
3. Matthews, Anthony A., et al. "Prospective benchmarking of an observational analysis in the SWEDEHEART registry against the REDUCE-AMI randomized trial." *European Journal of Epidemiology* 39.4 (2024): 349-361.

4. Immortal time bias: It is good that the authors used time-varying exposure to account for immortal time bias. However, the sensitivity analysis related to immortal bias seems incorrect. All the outcomes are at and after birth and after using time-varying exposure immortal time bias is not likely to be a problem in the current study but collider bias could be a problem due to the inclusion of only live births.

R: The sensitivity analysis to evaluate immortal time bias does not use time-varying exposure; in this analysis, the live birth is classified as exposed if the mother has an arbovirus infection anytime during pregnancy. We have now rewritten the sentence to make it clear.

“we classified the live-born as exposed since conception if their mother had an infection at any point during the pregnancy, i.e. not defining arbovirus infection as a time-varying exposure.”

And in the results section:

“In the analysis to assess immortal time bias, i.e. not using time-varying exposure, we observed the expected attenuation of results for all outcomes, notably for preterm and LBW, with a change in the direction of the association. (Supplementary Figure 3)”

RESULTS

Also, it would be informative to have a table with n (%) for the congenital anomalies probably by system (because of the small numbers) to evaluate the critical window of exposure because different systems/organs have different sensitive periods of development and the influence of teratogens is closely related to this.

R: This information is presented in supplementary Table 10 (previous supplementary table 9), breaking down by ICD-10 Chapter XVII and exposure status. Additionally, it is broken down by trimester for each arbovirus infection.

Please clarify how symptomatic chikungunya, dengue, and zika was defined.

R: The SINAN dataset (Notifiable Diseases Information System) only records information about symptomatic arbovirus infection. The exposure to Zika, dengue, and chikungunya was assessed through record linkage between the cohort and the Information System for Notifiable Diseases (SINAN). In this scenario any case linked between both system are only related to symptomatic disease

Table 1. Of the entire paper this is perhaps the only section that seems to be of low quality.

For example, it is unclear what the variables, days from chikungunya to conception date indicate. Is this for the current infection or days from previous infection? Similarly, it is unclear what the days for dengue and Zika viruses mean.

Please clarify what n (%) for clinical epidemiology and laboratory mean in Table 1 and why is it necessary in this table?

It is unclear why the two variables for Apgar score are separated by the congenital anomaly variable.

Also, it is difficult to comprehend what the first Apgar score means. Probably, it is the median (IQR) but there is no information provided.

It would be informative to provide gestational age at diagnosis (perhaps, the authors have reported it but since the table is poorly formatted it is not easy to understand the information provided).

Please modify this table to align with the results and also remove unnecessary information. It would be to organise the variables by certain characteristics.

R: We have now redesigned Table 1, incorporating the reviewer's points. The table now is separated into mother characteristics/ live-born characteristics/pregnancy arbovirus symptomatic infection / pre-conception arbovirus symptomatic infection. We also changed the formatting of the header of each variable and its categories, making it easier to read. We also changed “Gestational age on symptom onset” to “Gestational age (days) on arbovirus symptom onset”, making it clear the gestational age at diagnosis.

The “clinical epidemiology” and “laboratory” under “diagnosis criteria” in table 1 are the type of diagnosis criteria used to define the case as confirmed, as described in the methods:

“In Brazil, all suspected cases of chikungunya, dengue and Zika are recorded, and laboratory confirmation is required until a predefined incidence threshold is reached. Once this threshold is reached, diagnosis can be based on clinical-epidemiological criteria, considering typical symptoms occurring in the same area and timeframe as other confirmed cases and validation by epidemiological surveillance”

Figure 2: Good and informative figure

R: Thank you

Figure 3. Please add the variables adjusted in the models and also change the heading ‘Estimate’ to meaningful such as Adjusted hazard ratio (95%CI). The negative control does not seem to be robust for preterm birth because there was a significant association the magnitude of which is comparable to chikungunya. This may suggest that the use of a common negative control for all outcomes is not robust.

R: The table presents two effect measures (adjusted hazard ratios) for all birth outcomes and adjusted risk ratios for the neonatal outcome. We opted to use Estimate instead of Adjusted Hazard Ratio, as it would misrepresent the effect measure for neonatal death. Including “Adjusted Hazard Ratio/Risk Ratio” could lead readers to confuse birth outcomes with risk ratios. We have rephrased the footnote to make it clear:

“Figure 3. Overall effect estimates. Estimated adjusted hazard ratios (birth outcomes) and adjusted risk ratios (neonatal death), comparing groups exposed and unexposed to arbovirus infection during pregnancy by outcome. NCE = Negative control exposure.”

The same change was applied to Figure 4

Please add information on how many neonatal deaths were among preterm births and those who had congenital anomalies. This could be added to the text and not to the table.

R: We have now included this information at the end of the first paragraph of the results section.

“A total of 43,979 (0.6%) neonatal deaths occurred during the study period, among them 30,287 (68.9%) were from preterm births and 8,154 (18.5%) were from live births with a detected anomaly.”

Supplementary Table 9. It is not recommended to report numbers less than 5 or 10 in tables especially for rare diseases due to confidentiality reasons and statistical disclosure. Therefore, please combine categories appropriately or report only the percentages.

R: We have changed the table to report <5 (< %). Because the table is already aggregated by ICD-10 blocks, no further aggregation without loss of information is possible. The same correction was applied to Supplementary Table 11

Supplementary Figure 2 does not provide any useful information. Therefore, it can be deleted.

R: We have now excluded supplementary figure 2

DISCUSSION

Please note: Neonatal death is not a perinatal outcome.

A study limitation is collider bias because of the inclusion of only live births.

Because stillbirths and pregnancy losses were not included, the association for congenital anomalies is likely to be underestimated because a high proportion of pregnancy losses are congenital anomalies.

R: We thank the reviewer for the comment. We have discussed the point of collider bias, but using the term “live birth bias”. We have now clarified that in the discussion.

“Additionally, the absence of data on stillbirths or miscarriages can result in “live birth bias”, i.e. collider-stratification bias, potentially underestimating the adverse consequences of arbovirus infections on birth and neonatal outcomes if these infections increase the risk of stillbirths.”

MINOR COMMENT:

The use of the word, effects, suggests causality which cannot be inferred from most observational studies. Therefore, please change all instances of this to non-causal language.

R: We thank the reviewer for raising this important point about precision in terminology. We have now changed the wording in the text. E.g.:

“Few epidemiological studies have investigated the clinical **consequences** of arbovirus infection during pregnancy **on maternal, birth and neonatal outcomes**, and the existing literature has yielded mixed results.^{4,13–16} While the **consequences** of prenatal exposure to the Zika virus on offspring's neurological development are well documented,^{12,16,17}”

“In this nationwide study, we found that maternal chikungunya, dengue, and Zika during pregnancy were **associated** with increased risks of **adverse birth outcomes and neonatal death**. Maternal Zika had the strongest associations with most outcomes and was the only virus linked to increased risks of both small and large for gestational age (SGA and LGA) births. The **patterns of association** varied by trimester of infection:....”

P1)

This does not seem to be sufficient reason to exclude multiple births because there are methods to account for clustering due to multiple births such as multilevel modelling. However, if the authors insist on using data from singleton births, then please add this as a study limitation with citations because multiple birth is an important for preterm birth and congenital anomalies.

R: We have now included the restriction to singletons as an limitation in the discussion section:

“Eighth, we excluded multiple births (2% of the initial sample) because we are unable to identify which live births were from the same pregnancy. Since multiple births are linked to a higher risk of adverse outcomes, this could bias our analysis if most were from unexposed women. However, because our exposure (arbovirus infection) occurs after conception, it is very unlikely that there is any association between the exposure and the occurrence of multiple births. In this scenario, with rare multiple births and no association between multiple births and the exposure, estimates from approaches that restrict on singletons will yield similar results to those evaluated at the infant level.⁴⁰”

40. Brown, J. P., Yland, J. J., Williams, P. L., Huybrechts, K. F., & Hernández-Díaz, S. (2025). Accounting for Twins and Other Multiple Births in Perinatal Studies of Live Births Conducted Using Healthcare Administration Data. *Epidemiology*, 36(2), 165-173.

P2)

Thank you for the information provided regarding the use of the indicator method. However, the purpose of the indicator method analysis is still unclear. In this study, analysis based on multiple imputation is the primary analysis. Therefore, one may expect to compare the results from these analyses with a complete case analysis to assess if missing data could influence the results. However, it is unclear why it was necessary to compare the results from MI with indicator method. Also, could you please describe the ‘certain circumstances’ (missing indicator method can provide valid inferences with minimal computational cost), with respect to this study.

I may have missed it but there seems to be no response to the suggestion: It would also be informative to compare the characteristics of women with and without missing data and to provide the extent of missing values.

R: In the recent literature, there are different scenarios in which the Missing Indicator Method (MIM) could result in reliable inference. Song et al derives an analytical formula to quantify the % of bias in the presence of missing, based in 5 parameters (the prevalence of the exposure, the prevalence of the covariate, the proportion of missingness, the effect of the covariate on the outcome, and the association between the exposure and the covariate), showing that if the variable with missing value is only associated with the outcome or no association, it won't biased the results. In addition, in their parameter evaluation space, only in cases with higher proportion of missing (=50%) or strong association between the variable with missing and exposure (RR=5 or 1/5) between 5 and 10% of parameter spaces show bias greater than 5% and between 3 and 5% greater than 10%. Lastly, Zhuchkova and Rotmistrov evaluate the applicability of each method through simulations, with the result that MIM (and complete case analysis) produces unbiased estimates under MCAR (missing complete at random) and MNAR (missing not at random), while Multiple Imputation has advantage under MAR, with similar results under MCAR, but biased results under MNAR. According to Blake, the MIM will be valid if (i) there is no unmeasured confounding within missingness patterns; (ii) either confounder values of patients with missing data are conditionally independent of treatment assignment, or these missing confounder values are conditionally independent of the outcome; and (iii) the effect of fully observed confounders on the outcome is the same for all missingness patterns. In their simulations, they further evaluate that in scenarios with weak

violation of the assumptions (conditionally independent of the outcome or treatment) the resulting bias is similar to no violation.

Considering the increased use of administrative databases and electronic health records with similar levels of missing data to our study (less than 6% in each individual variable), and the exponential computing costs in complex analysis in large datasets (measured in millions) (such as time-varying covariates, standardization/parametric g-formula, analysis requiring bootstrapping) when conducted in multiple imputed datasets. The comparison of both approaches in our articles provides additional examples of similar results under the three approaches. As outlined in the methods (“Each analysis considers a different type of underlying mechanism for the missing data (at random, completely at random or not at random)”, each method theoretically would work better with one missing pattern MI (MAR), MIM (MNAR), CCA (MCAR). Our article provides the results of the three approaches in a large observational database, generating additional resources for further studies on the topic. Lastly, we do not include a table comparing the total of individuals with missing (695,829 – 10%) as this comparison is only relevant for the complete case analysis; however in table 1 and supplementary table 1 all variables with missing values are included and showed the relative percentage in each exposure group.

P3)

3. The negative control does not seem to be robust for preterm birth because there was a significant association the magnitude of which is comparable to chikungunya. This may suggest that the use of a common negative control for all outcomes is not robust

R: We included the same NCE, by the rationale that previous arbovirus infection (accounting for 6 month washout) wouldn't be associated with any outcome related to the birth. The use of the same NCE in all outcomes allows for direct comparison of the amount of possible bias due to residual confounding of the main exposure in the different outcomes. While the results of the NCE group for preterm are similar to the overall effect of dengue (1.04 versus 1.07), it should be noted that the dengue effect is not homogenous across the trimesters, as indicated by the Cochran test (supplementary table 2). Dengue in the third trimester had an effect of 1.21. The presence of significant results in NCE are expected, as it is a tool to detect residual confounding {1}, we have rephrased the text regarding NCE in the results section:

“Our negative control exposure analysis, using pre-conception infection, suggested minimal residual confounding, as the estimates were generally closer to the null than results for the main analyses (specially by trimester), with small associations detected for preterm birth, LBW and LGA (Figure 3).”

Of note this was also included in the limitation paragraph:

“To address this, we used a negative control exposure (infection before pregnancy) to test for spurious associations, and in this analysis, we detected residual confounding”

{1} Lipsitch, Marc, Eric Tchetgen Tchetgen, and Ted Cohen. "Negative controls: a tool for detecting confounding and bias in observational studies." *Epidemiology* 21.3 (2010): 383-388.